# Population Pharmacokinetics of Orally Administered Clindamycin to Treat Prosthetic Joint Infections: A Prospective Study

**DOI:** 10.3390/antibiotics11111462

**Published:** 2022-10-23

**Authors:** Léo Mimram, Sophie Magréault, Younes Kerroumi, Dominique Salmon, Benjamin Kably, Simon Marmor, Anne-Sophie Jannot, Vincent Jullien, Valérie Zeller

**Affiliations:** 1Unité Fonctionnelle de Pharmacologie, Groupe Hospitalier Paris Seine Saint-Denis, 93143 Bondy, France; 2IAME UMR 1137, Inserm and Paris Diderot University, Team Biostatistic Modelling, Clinical Investigation and Pharmacometrics in Infectious Diseases, 75870 Paris, France; 3Centre de Référence des Infections Ostéo-Articulaires Complexes (CRIOAC), Groupe Hospitalier Diaconesses–Croix Saint-Simon, 75020 Paris, France; 4Service de Médecine Interne, Hôpital Cochin, Assistance Publique–Hôpitaux de Paris (APHP), 75014 Paris, France; 5Service de Pharmacologie DMU BioPhyGen, Hôpital Européen Georges-Pompidou, APHP, 75015 Paris, France; 6Service d’Informatique Médicale, Biostatistiques et Santé Publique, Hôpital Européen Georges-Pompidou, APHP, 75015 Paris, France; 7Service de Médecine Interne et Infectiologie, Groupe Hospitalier Diaconesses–Croix Saint-Simon, 75020 Paris, France

**Keywords:** clindamycin, population pharmacokinetics, pharmacokinetic parameter, Monte-Carlo simulations

## Abstract

A population PK model of clindamycin orally administered to patients with prosthetic joint infections (PJIs) was developed using NONMEM 7.5. Monte-Carlo simulations were run to determine the probability of obtaining bone clindamycin concentrations equal to at least the MIC or four times the MIC for several MIC values and dosing regimens. One hundred and forty plasma concentrations prospectively obtained from 20 patients with PJIs were used. A one-compartment model with first-order absorption and elimination appropriately described the data. Mean PK-parameter estimates (F being the bioavailability) were: apparent clearance, CL/F = 23 L/h, apparent distribution volume, V/F = 103 l and absorption rate constant, Ka = 3.53/h, with respective interindividual variabilities (coefficients of variation) of 14.4%, 8.2% and 59.6%. Neither goodness-of-fit curves nor visual predictive checks indicated bias. The currently recommended 600 mg q8h regimen provided a high probability of obtaining concentrations equal to at least the MIC, except for MIC ≥ the clinical breakpoint for *Staphylococcus* spp. (0.25 mg/L). For such MIC values, higher daily doses and q6h regimens could be considered.

## 1. Introduction

Clindamycin is a lincosamide antibiotic known for its good penetration into bone, and its activity against Gram-positive aerobic and anaerobic species. The main pathogens involved in prosthetic joint infections (PJIs) are Gram-positive cocci with 65.2% of those infections caused by *Staphylococcus* spp., followed by 20.4% Enterobacteriaceae and 9% *Streptococcus* spp. [1]. Therefore, clindamycin is widely used to treat PJIs. However, only few studies have investigated clindamycin pharmacokinetics (PK) in this context. Two studies described the peak and trough concentrations that were obtained during oral or intravenous treatment and the impact of rifampin on them [2]. Another study also retrospectively evaluated the drug interaction with rifampin when clindamycin was administered as continuous infusion [3]. A population analysis after oral or intravenous clindamycin administration, alone or in combination with rifampicin, was performed, but based on therapeutic drug-monitoring (TDM) data [4]. The only prospective study published to date was our group’s non-compartmental PK analysis, which demonstrated the major influence of clindamycin-administration route on the magnitude of its interaction with rifampin [5], a phenomenon that could not be identified in the retrospective, population-PK analysis cited above. All these previous studies aimed at evaluating the PK interaction between clindamycin and rifampin. Only the previously published population PK model was used to evaluate clindamycin dosing regimens. However, this model, based on a retrospective sparse sampling design, could not identify rifampin as a significant covariate, whereas it included both patients receiving and not receiving concomitant rifampin, and did not take into account the MIC distribution of *Staphylococcus* and *Streptococcus* spp. to evaluate clindamycin dose [4]. Consequently, a population-PK study conducted with prospectively-acquired data could be of interest to estimate robust clindamycin-PK parameters, their interindividual variability, and to evaluate clindamycin-dosing regimens, taking into account the microbiological variability.

We developed a population-PK model of clindamycin after oral intake in patients with PJIs, using the data from our previous study, to evaluate its interindividual PK variability and identify optimal oral dosing regimens to achieve target concentrations.

## 2. Results

### 2.1. Patients

Twenty patients, all treated for a PJI, were included. Their characteristics are given in Table 1. One hundred and forty plasma clindamycin concentrations, ranging between 0.4 and 13.5 mg/L, were available.

### 2.2. PK Parameters

A one-compartment model with first-order absorption and elimination and mixed residual error best described the data.

During the forward covariate-inclusion process, none had a significant impact on the objective function. Therefore, no covariate was included in the model.

Mean PK parameters and their relative standard errors, combined with the bootstrap results, are reported in Table 2.

η-shrinkages for CL, volume (V) and absorption rate constant (Ka) were 0%, 8% and 22%, respectively, and ε-shrinkage was 19%. 

No bias was observed on the goodness-of-fit curves (Figure 1) or on the pvcVPC (Figure 2).

### 2.3. Dosing-Regimen Evaluation

Monte-Carlo–simulation results are shown in Table 3. The usual 450–600 mg q8h doses could provide probability of target attainment (PTA) ≥ 90% only for the lowest PK/PD target and for MICs inferior to the clinical breakpoint for *Staphylococcus* spp. (0.25 mg/L). For this MIC value, only q6 regimens and the 900 mg q8h regimen provided a satisfying PTA. For the MIC = 0.5 mg/L corresponding to the clinical breakpoint for *Streptococcus* spp., only the 900 mg q6h regimen allowed the achievement of a PTA ≥ 90%. PTAs ≥ 90% for the highest PK/PD target could be obtained only for MICs ≤ 0.125 mg/L and with the 900 mg q8h or the q6h regimens.

## 3. Discussion

The 140 plasma clindamycin concentrations obtained from 20 patients in a controlled environment allowed us to build our model on robust data.

A linear one-compartment model with first-order absorption appropriately described the data. The same structural model was used in the previously published population PK study [4]. The mean CL/F and V/F obtained in that study (17.3 L/h and 75.6 L, respectively) were slightly lower than our values of 23 L/h and 100 L. Our mean CL/F estimate is, nevertheless, similar to the values previously published for AIDS patients (24.8 L/h) and healthy women and men (23.5–26.5 L/h); mean V/F for healthy subjects (120–130 L) was slightly higher than our mean estimate [6,7].

We did not identify a relevant covariate able to explain the interindividual clindamycin-PK variability, which likely reflects the somehow low number of patients compared to the number of investigated covariates and the relative closeness of the distributions of the covariates investigated, since patients with abnormal hepatic or kidney functions were excluded, as were subjects with too low or too high bodyweights, or subjects with severe infections. Hence, further studies, with a higher number of patients, are needed to investigate those additional factors.

To our knowledge, no PK/PD target was defined to date for clindamycin in patients with PJIs. Based on its time-dependent PK/PD profile, obtaining a concentration at the site of infection at least equal to the MIC could be a reasonable objective (C_trough_ ≥ 2 × MIC). According to our simulations, the usually prescribed 600 mg q 8h regimen allows the achievement of this target with a high probability, except for MICs equal to the clinical breakpoints for *Staphylococcus* and *Streptococcus* (i.e., 0.25 and 0.5 mg/L). However, according to EUCAST data, only 0.3% of Staphylococcus strains and 3.6% of Streptococcus strains susceptible to clindamycin have a MIC value ≥ 0.25 mg/L. Consequently, this regimen could appear satisfying. Indeed, taking into account the EUCAST MIC distributions for susceptible strains, the PK/PD target would be achieved for 95% of *Staphylococcus* and for 96% of *Streptococcus* strains with the 600 mg q8h dose. On the other hand, it is interesting to consider that this global PTA seems superior to the probability of clinical efficacy that was described in several observational studies. For instance, in three French cohorts of patients with bone and joints infections (BJIs), clindamycin-based treatments were characterised by clinical success rates of 83% (71% when clindamycin was used as monotherapy), between 67 and 85% for infections due to erythromycin-resistant *Staphylococcus* strains, and 91% [8,9,10]. This gap with the PTA we obtained for the low PK/PD target could suggest that the trough clindamycin concentration at the site of infection should be greater than the MIC, or that the trough plasma concentration of clindamycin should be greater than twice the MIC of the causative bacteria. By comparison, the global PTA for the same dose and the highest PK/PD target (C_trough_ ≥ 8 × MIC) given by our simulations is around 73% for *Staphylococcus* and 79% for *Streptococcus* strains. Our simulations also suggest that in case of MIC values close to the clinical breakpoint for *Staphylococcus*, higher doses and more particularly q6h regimens could be of interest. However, the tolerance of such doses should be validated.

We found no study providing simultaneously efficacy, pharmacological and microbiological data in patients with PJIs or BJIs receiving clindamycin-based therapy. Such data would be important to investigate whether clindamycin concentration should be individually monitored, concomitantly with MIC determination, in order to improve clindamycin efficacy.

TDM is now widely acknowledged as an important tool to optimise antibiotic treatment [11,12]. However, the TDM contribution was more particularly highlighted for ICU patients and other antibiotic classes, such as aminoglycosides, glycopeptides, β-lactams and fluoroquinolones [13,14,15]. It can nonetheless be noted that the probable need for TDM of patients with PJIs was previously suggested for vancomycin, highlighting the importance of dose optimisation in this context [14]. This result also suggests that the plasma concentration of an antibiotic may be used as a surrogate marker of the concentration at the site of infection. Measuring clindamycin concentration in bone samples was not possible in the present study. We therefore used a previously published diffusion ratio of 50% to derive the bone concentration of clindamycin from its plasma concentration. This 50% ratio represents approximately the mean value obtained from previously published studies, but the important between-study variability of this ratio makes our choice questionable [16,17]. The measurement of the concentration of a drug into bone raises many issues, including sample availability, the choice of the matrix to be analysed (micro-dialysate or bone homogenate) and the analytical method [17]. Because of all these pitfalls, studies evaluating the potential interest of clindamycin TDM based on its plasma concentrations now seem warranted, especially for patients with PJIs/BJIs.

In conclusion, the results of this novel population clindamycin-PK model of patients with PJIs based on a prospective protocol and Monte-Carlo simulations supported the appropriateness of the current 600 mg q8h regimen in most cases, but higher daily doses divided into four intakes seem necessary for MIC values close to the EUCAST clinical breakpoints. Furthermore, the broad variability of the trough concentrations that was observed suggested the credible contribution of clindamycin TDM. Further studies are needed to investigate that possibility.

## 4. Materials and Methods

### 4.1. Ethics

The previous study from which we obtained our data [5] was approved by the Île-de-France Ethics Committee (Comité de Protection des Personnes, CPP Île-de France VI, Groupe Pitié–Salpêtrière, no. 142–14). In accordance with French legislation, written consent was obtained from all participants.

### 4.2. Patients

Our earlier prospective study was undertaken to describe and characterise the impact of the clindamycin-administration route on its PK interaction with rifampicin (ClinicalTrials.gov, accessed on 9 September 2022, Identifier: NCT 02629770) [5]. Only the data obtained from the patients not prescribed rifampicin were used for the present population-PK analysis.

This multicenter study was conducted in the Referral Center for Bone and Joint Infections [18] of Diaconesses-Croix Saint-Simon Hospital (Paris, France) and Cochin Hospital (Paris, France) from December 2015 to November 2019.

The inclusion criteria were patients at least 18 years old hospitalised for a chronic PJI, evolving for >4 weeks and caused by clindamycin- and rifampicin-susceptible staphylococci, streptococci or anaerobic bacteria.

Patients were excluded if they had one or more of the following criteria: allergy to clindamycin and/or rifampicin, hepatocellular insufficiency, cirrhosis, creatinine clearance <30 mL/min, severe sepsis or septic shock, porphyria, congenital galactosaemia, concomitant glucose-and-galactose–malabsorption syndrome, lactase deficit, fructose or glucose intolerance, sucrase–isomaltase deficit, major cognitive disorders, body weight >100 or <50 kg, concomitant therapies able to induce or inhibit cytochrome P450 3A4/A5, pregnancy, lactation, or women using estrogen–progestin contraceptives.

### 4.3. Treatment

After a first week of usual post-operative, intravenous (IV) antibiotics, patients received IV clindamycin for 2–5 weeks, followed by 3–6 weeks of oral clindamycin. The oral clindamycin dose was determined according to weight: 750 mg q8h for patients <80 kg or 900 mg q8h for patients ≥80 kg; it could be raised to 1200 mg q8h for patients >95 kg.

### 4.4. PK Sampling

After 2 weeks of oral administration, seven blood specimens were drawn between two oral intakes at 0, 0.5, 1, 2, 4, 6 and 8 h.

### 4.5. Analytical Methods

Each of those seven blood samples were immediately centrifuged and stored at −20 °C until analysis. Liquid chromatography–mass spectrometry determined plasma clindamycin concentrations with a lower limit of quantification of 0.09 mg/L.

### 4.6. PK Models

Concentrations–time data were analyzed using the non-linear mixed-effects modelling program NONMEM 7.5, combined with the Pirana graphical user interface [19]. One- and two-compartment models, with zero- and first-order absorption processes were investigated. Interindividual variability was described by assuming that individual parameters arise from a multivariate log-normal distribution with mean vector and variance–covariance matrix to be estimated. Finally, additive, exponential and mixed residual error models were tested.

### 4.7. Covariates

The following covariates were investigated: sex, age, weight, height, bilirubin, creatinine and alanine aminotransferase. The influence of continuous covariates on the PK parameters was systematically tested using a generalised model, according to the following equations:CL = TV(CL/F) × (COV/median COV) ^θcov^(i)
where TV(CL/F) was the typical value of the apparent clearance, F the bioavailability, for a patient with the median covariate (COV) value, and θcov was the corresponding influential factor.

Categorical covariates (sex), were evaluated according to the following equation:CL = TV(CL/F) × θ_COV_(ii)
where θ_COV_ was estimated for patients with the covariate (combined drug and male) and was otherwise fixed at 1.

The significance of a PK-parameter–covariate relationship was assessed using the chi-square test of the difference between the objective functions of the basic model (without the covariate) and the model with the covariate. A covariate was retained in the model if it produced a minimum 4-unit decrease of the objective function (*p* = 0.05, 1 degree of freedom) and its effect was biologically plausible. An intermediate multivariate model that included the significant covariates was then obtained. The covariates were retained in the final multivariate model if their deletion from the intermediate model led to a 7-point increase of the objective function (*p* = 0.01, 1 degree of freedom). At each step, goodness-of-fit was evaluated using a graph of the conditional weighted residuals versus the predicted concentration or the hour post-dose intake. 

### 4.8. Model Evaluation

The accuracy and robustness of the final population model were assessed by a bootstrap resampling technique (resampling repeated 500 times) and by a prediction- and variability-corrected visual predictive check (pvcVPC) [20]. Lack of bias was also evaluated by inspection of the conditional weighted residuals with respect to time after dose intake and population predictions. Goodness-of-fit was also visualised on graphs of observed versus individual- or population-predicted concentrations. Bootstrap and pvcVPC were obtained using PsN combined with Pirana [19].

### 4.9. Dosing-Regimen Simulations

The final clindamycin population-PK model was used to evaluate the probability of reaching the target clindamycin-trough concentrations for different dosage regimens. Based on its bone penetration ratio of 50% [16] and its time-dependent PK/PD profile, a minimal target trough plasma concentration equal to twice the MIC would allow the achievement of a bone clindamycin concentration > MIC over the entire dosing interval. Another more stringent target, consisting in a trough plasma clindamycin concentration/MIC ratio of at least 8, corresponding to a bone clindamycin concentration MIC ratio of at least 4, was also considered, as this target is suggested for other time-dependent antibiotics such as beta-lactams [12]. Monte Carlo simulations were performed using the final model to determine the probability to attain these two targets for seven dosing regimens (450 mg q6h, 600 mg q8h, 600 mg q6h, 750 mg q8h, 750 mg q6h, 900 mg q8h, 900 mg q6h) and MIC values ranging between 0.0625 and 0.5 mg/L). Five hundred patients were simulated for each regimen/MIC combination.

Furthermore, global PTAs of both targets for the seven dosing regimens were estimated using the EUCAST MIC distributions for susceptible Staphylococcus and Streptococcus strains (i.e., by considering only MIC values ≤ to their respective clinical breakpoints). To our knowledge, no clindamycin concentration–adverse event relationship has been reported to date. Consequently, the upper limit of target clindamycin-trough concentrations was arbitrarily set at 5 mg/L, as this corresponds to the lowest maximal concentration observed in this population [5].

## Figures and Tables

**Figure 1 antibiotics-11-01462-f001:**
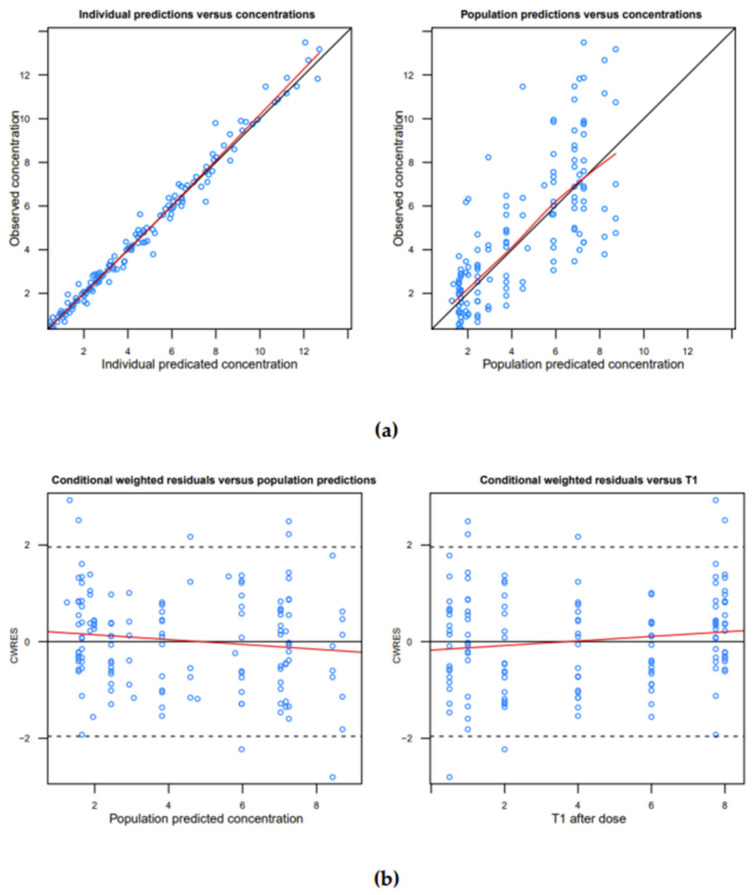
(**a**) Observed versus (left) individual-predicted or population-predicted (right) clindamycin concentrations (mg/L); or (**b**) conditional weighted residuals (CWRES) versus (left) population-predicted concentration or (right) hour post-dose intake (T1 after dose).

**Figure 2 antibiotics-11-01462-f002:**
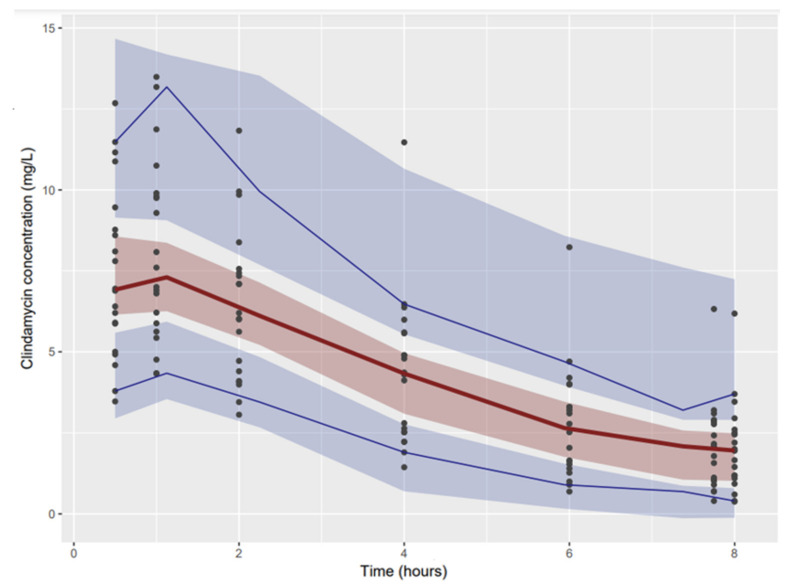
Prediction- and variability-corrected visual predictive (pvcVPC) for clindamycin concentrations after oral intake using the final model. The lower blue, red and upper blue lines show, respectively, the 10th, 50th and 90th percentiles obtained with observed data. The lower blue, red and upper blue areas represent the 90% confidence intervals obtained with simulated concentrations around their respective 10th, 50th and 90th percentiles.

**Table 1 antibiotics-11-01462-t001:** Characteristics of the 20 patients treated with clindamycin for chronic prosthetic joint infections.

Characteristic	Value
Males/females, *n*	13/7
Body weight (kg)	76.3 (14.4)
Age (years)	66.5 (15.9)
Prothrombin time (%)	85.7 (12.9)
Total proteins (g/L)	72.8 (6.15)
Albumin (g/L)	37.6 (5.11)
Aspartate aminotransferase (IU/L)	16.0 (4.41)
Alanine aminotransferase (IU/L)	14.6 (6.10)
Bilirubin (µmol/L)	6.83 (3.49)
Creatinine (µmol/L)	73.6 (17.2)
C-reactive protein (mg/L)	47.6 (79.1)
Leucocytes (G/L)	8.19 (3.55)
Infection localization, *n*	
Shoulder arthroplasty	3
Hip arthroplasty	14
Knee arthroplasty	3

Values are mean (standard deviation), unless stated otherwise. CRP, C-reactive protein.

**Table 2 antibiotics-11-01462-t002:** Estimated mean population pharmacokinetic parameters and relative standard errors obtained with the original dataset and their corresponding bootstrap values.

	Original Dataset		Bootstrap	
Parameter	Estimate	RSE	Estimate	RSE
Structural model				
CL/F (L/h)	23.00	8.7%	23.05	9.2%
V/F (L)	103.00	8.0%	102.1	7.6%
Ka (/h)	3.53	22.4%	3.71	29.6%
Interindividual variability				
ω^2^ CL/F	0.14	24.7%	0.136	25.9%
ω^2^ V/F	0.08	28.2%	0.078	28.9%
ω^2^ Ka	0.60	46.3%	0.63	66%
Residual error				
Proportional error Additive error	0.00976 0.0801	36.0% 49.0%	0.009240 0.0827	34.4% 43.2%

RSE: relative standard error; F: bioavailability: CL/F: apparent clearance; V/F: apparent distribution volume; Ka: absorption rate constant; ω^2^ CL/F: interindividual variability of CL/F; ω^2^ V/F: interindividual variability of V/F; ω^2^ Ka: interindividual variability of Ka.

**Table 3 antibiotics-11-01462-t003:** Simulated clindamycin trough concentrations (median and 90% non-parametric confidence interval, CI) and probability of achieving the PK/PD targets of trough concentration (Ctrough) ≥ 2 and 8 × MIC per dosing regimen.

Dosing Regimen	450 Mg q6h	600 Mg q8h	600 Mg q6h	750 Mg q8h	750 Mg q6h	900 Mg q8h	900 Mg q6h
Median 90% CI	1.73 0.42–4.10	1.31 0.21–4.67	2.10 0.41–5.51	1.51 0.19–4.92	2.81 0.63–6.73	1.92 0.31–5.89	3.38 0.71–8.55
Probability of reaching C_trough_ ≥ 2 × MIC							
MIC = 0.5 mg/L	75%	64%	81%	70%	89%	76%	92%
MIC = 0.25 mg/L	93%	86%	94%	85%	97%	91%	98%
MIC = 0.125 mg/L	97%	93%	97%	93%	99%	97%	99%
MIC = 0.0625 mg/L	99%	97%	99%	97%	100%	99%	100%
MIC = 0.03125 mg/L	100%	99%	100%	98%	100%	99%	100%
Global PTA for *Staphylococcus*	98%	95%	98%	95%	100%	98%	100%
Global PTA for *Streptococcus*	98%	96%	99%	95%	99%	98%	100%
Probability of reaching C_trough_ ≥ 8 × MIC							
MIC = 0.5 mg/L	6%	8%	16%	9%	29%	15%	38%
MIC = 0.25 mg/L	39%	30%	53%	37%	68%	48%	74%
MIC = 0.125 mg/L	75%	64%	81%	70%	89%	76%	92%
MIC = 0.0625 mg/L	93%	86%	94%	85%	97%	91%	98%
MIC = 0.03125 mg/L	97%	93%	97%	93%	99%	97%	99%
Global PTA for *Staphylococcus*	82%	73%	86%	76%	92%	82%	95%
Global PTA for *Streptococcus*	87%	79%	90%	80%	94%	86%	96%
Probability to reach the upper limit concentration							
>5 mg/L	2%	3%	8%	5%	18%	9%	25%

## Data Availability

Not applicable.

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
