# Peer review of "Population Pharmacokinetics of Orally Administered Clindamycin to Treat Prosthetic Joint Infections: A Prospective Study"

_antibiotics, 2022, doi:10.3390/antibiotics11111462_

Round 1
Reviewer 1 Report
This is a very good study with sufficient information, and it is well presented. But I would like to know, that why authors did not measure the MIC distributions for susceptible Staphylococcus and Streptococcus strains from the location of study or hospital? Why did authors not discuss about the risk of toxicity based on new dosing regimens? Additionally, authors did not conduct microdilution study to verify the clindamycin concentration in bone tissue or skin and soft tissues.
Author Response
a- This is a very good study with sufficient information, and it is well presented. But I would like to know, that why authors did not measure the MIC distributions for susceptible Staphylococcus and Streptococcus strains from the location of study or hospital?
We thank the reviewer for this kind comment. MIC are not routinely measured in our institution, only inhibition diameters are determined. However, we used the MIC distributions provided by EUCAST which, to our opinion, brings robustness to our results.
b- Why did authors not discuss about the risk of toxicity based on new dosing regimens?
We thank the reviewer for this important comment and added the following sentence in the discussion, Line 171-172: “Our simulations also suggest that in case of MIC values close to the clinical breakpoint for Staphylococcus, higher doses, and more particularly q6h regimens could be of interest. However, the tolerance of such doses should be validated”
c- Additionally, authors did not conduct microdilution study to verify the clindamycin concentration in bone tissue or skin and soft tissues.
We acknowledge this is a limitation of the study. Such an analysis was not possible in our study as bone samples could not be obtained since there was no surgery while the patients were on treatment. Consequently we used a diffusion ratio of 50 % to derive clindamycin bone concentration from our simulations. However we acknowledge the choice of this 50 % ratio is arbitrary and does not reflect the important difference in its value that can be observed from a study to another. Until an easy method to measure drug concentration into bone is available, we believe studies investigating the usefulness of plasma concentrations are warranted.
To reflect this we added the following paragraph to the discussion (Line 183-193):
“This result also suggests that the plasma concentration of an antibiotic may be used as a surrogate marker of the concentration at the site of infection. Measuring clindamycin concentration into bone samples was not possible in the present study. We therefore used a previously published diffusion ratio of 50 % to derive the bone concentration of clindamycin from its plasma concentration. This 50 % ratio represents approximately the mean value obtained from previously published studies, but the important between-study variability of this ratio makes our choice questionable [15,16]. The measurement of the concentration of a drug into bone raises a lot of issues, including sample availability, the choice of the matrix to be analysed (microdialysate or bone homogenate), and the analytical method [16]. Because of all these pitfalls, studies evaluating the potential interest of clindamycin TDM based on its plasma concentrations now seem warranted, especially for patients with PJIs”.
Reviewer 2 Report
The current study addresses a very important topic, which is antibiotic pharmacokinetics, since it directly affects their efficacy and safety. The study’s methods are quite well described, and the data has been thoroughly analyzed. Nevertheless, the analyzed samples were obtained from 20 patients only, which is a very small population, not enabling to make generalized and, consequently, more useful conclusions. Hence, I do not advise the acceptance of this manuscript for publication in Antibiotics, given the high standard that this journal requires.
Author Response
As it was discussed in the manuscript, our estimated PK parameter values are in the range of previously published results, evidencing the reliability of our results. However, we acknowledge that a higher number of patients would be necessary to identify the possibly relevant covariates. Consequently, we modified a sentence of the discussion as follows (Line 145): “Hence, further studies, with a higher number of patients, are needed to investigate those additional factors”
Reviewer 3 Report
Dear author, this study provide an evidence based of clinical efficacy of orally administered clindamycin used in the treatment of prosthetic joint infections. However, there is still scope of further improvement in the manuscript. Therefore, I recommend to accept the manuscript after a minor revision.
Here are some comments related to the manuscripts:
1. Improve the introduction and discuss a few latest studies related to the work.
2. Conclusion should be more impressive and detailed.
Regards!
Author Response
We thank the reviewer for his comment. The introduction and discussion were improved after adding new references. More details were provided to the conclusion.
Reviewer 4 Report
This paper is interesting and worth publishing. The following aspects should be considered, in our view, by the authors:
Line 221: the Nonmem software name is written wrongly (NONMEN should be corrected to NONMEM)
Lines 229-258: the stepwise (forward) approach to building the model could be challenged (as stepwise regression is fraught with methodological issues and uncertainties), but it can be considered reasonable. The validation approach, instead, is rather vague to me (it is not clear to me how the pvcVPC works). Moreoever, I think that a (double) cross-validation procedure would allow one a close-to-true validation of the model. At the end of the day there are 140 observations coming out from only 20 patients (there is autocorrelation among those 140 observations) and the number of variables evaluated is 7. This should be acknowledged as a limitation. Could the fact that no covariate was significant be the mere result of low power? Could also the authors explore combining these data with others from the literature in an attempt to increase the sample size?
The PTA abbreviation is used throughout the paper with no definition. It could be defined on line 101, with an appropriate change of words order (i.e. there replace “target attainment probabilities” with “probability of target attainment (PTA)”.
Author Response
This paper is interesting and worth publishing. The following aspects should be considered, in our view, by the authors:
a- Line 221: the Nonmem software name is written wrongly (NONMEN should be corrected to NONMEM)
Answer : We thank the reviewer for pointing out this unfortunate mistake and corrected it.
b- Lines 229-258: the stepwise (forward) approach to building the model could be challenged (as stepwise regression is fraught with methodological issues and uncertainties), but it can be considered reasonable. The validation approach, instead, is rather vague to me (it is not clear to me how the pvcVPC works). Moreoever, I think that a (double) cross-validation procedure would allow one a close-to-true validation of the model. At the end of the day there are 140 observations coming out from only 20 patients (there is autocorrelation among those 140 observations) and the number of variables evaluated is 7. This should be acknowledged as a limitation. Could the fact that no covariate was significant be the mere result of low power? Could also the authors explore combining these data with others from the literature in an attempt to increase the sample size?
Answer: We thank the reviewer for raising these important points. pvcVPC can be easily implemented using the PsN scripts and the Pirana interface by selecting the “predcorr” and “varcorr” options before running the VPC. This is now explained in the method section (Line 245 and lines 283-284). We performed a supplemental internal validation consisting in a bootstrap resampling technique (resampling repeated 500 times). The results are similar to the estimated values obtained with the original dataset, with the exception of the RSE of the IIV of ka which is higher with the boostrap procedure (66% instead of 46 %). These results were included in the Results section
The somehow low sample size, combined to the homogeneous profile of the included patients in terms of covariate values, is certainly the main reason why we were not able to identify any significant covariate. All these aspects were added to the discussion (Line 141). Last, it was not possible to use the data of the previously published population PK model.
c-The PTA abbreviation is used throughout the paper with no definition. It could be defined on line 101, with an appropriate change of words order (i.e. there replace “target attainment probabilities” with “probability of target attainment (PTA)”.
Answer: We thank the reviewer for pointing out this problem. The sentence was corrected accordingly
Round 2
Reviewer 2 Report
Given the ackowledgement by the authors of the study's limitations, and given that they added a statement regarding this in the manuscript, I am confortable with publication as long as the other reviewers and the editor agree.